# Development and real-life use assessment of a self-management smartphone application for patients with inflammatory arthritis. A user-centred step-by-step approach

Catherine Beauvais[1]*, Thao Pham[2], Guillaume Montagu[3], Sophie Gleizes[3,4], Francesco Madrisotti[3,5], Alexandre Lafourcade[6,7], Céline Vidal[8], Guillaume Dervin[9], Pauline Baudard[10], Sandra Desouches[1], Florence Tubach[6,7], Julian Le Calvez[11], Marie de Quatrebarbes[11], Delphine Lafarge[12], Laurent Grange[13], Françoise Alliot-Launois[13], Henri Jeantet[11], Marie Antignac[14,15], Sonia Tropé[16], Ludovic Besset[11], Jérémie Sellam[1,17], on behalf of Therapeutic patient education group of the French Society of Rheumatology and Club Rhumatismes et Inflammations[¶]

1 Service de Rhumatologie, Hôpital Saint Antoine, Assistance Publique Hôpitaux de Paris (AP-HP), Sorbonne Université, Paris, France, 2 Service de Rhumatologie, Hôpital Sainte Marguerite, Aix Marseille Univ, Assistance Publique Hôpitaux de Paris (APHM), Marseille, France, 3 Département de Recherche, Unknowns, Conseil en Stratégie et Innovation, Paris, France, 4 Laboratoire d'Anthropologie Sociale (LAS), Collège de France/EHESS/EPHE, Paris, France, 5 Centre de National de la Recherche Scientifique (CNRS), Laboratoire CERMES3 (CNRS-EHESS-INSERM), Université de Paris, Paris, France, 6 Institut Pierre Louis d'Epidémiologie et de Santé Publique, Sorbonne Université, INSERM, Paris, France, 7 Département de Santé Publique, Centre de Pharmacoépidémiologie (Cephepi), CIC-1422, Hôpital Pitié Salpêtrière, Sorbonne Université, APHP, Paris, France, 8 Clinique Beau Soleil, Montpellier, France, 9 Clinique Juge, Institut Médical Sport Santé Marseille, Marseille, France, 10 Service de Rhumatologie, Centre Hospitalo-Universitaire Caen, Caen, France, 11 Unknowns, Conseil en Stratégie et Innovation, Paris, France, 12 Association France Spondyloarthrite (AFS), Tulle, France, 13 AFLAR Association Française de Lutte Anti rhumatismale (AFLAR), Paris, France, 14 Service de Pharmacie, Hôpital Saint Antoine, Assistance Publique Hôpitaux de Paris (AP-HP), Sorbonne Université, Paris, France, 15 Equipe d'Épidémiologie Intégrative, INSERM U 970, PARCC, Paris, France, 16 Association Nationale de Défense Contre l'Arthrite Rhumatoïde (ANDAR), Paris, France, 17 Centre de Recherche Saint-Antoine (CRSA), INSERM UMRS_938, FHU PaCeMM, Paris, France

¶ Membership of Therapeutic patient education group of the French Society of Rheumatology and Club Rhumatismes et Inflammations are provided in the Acknowledgments.
* catherine.beauvais@aphp.fr

## Abstract

### Background

Mobile health applications (apps) are increasing in interest to enhance patient self-management. Few apps are actually used by patients and have been developed for patients with inflammatory arthritis (IA) treated with disease-modifying anti-rheumatic drugs which use entails risk of adverse effects such as infections.

### Objective

To develop Hiboot, a self-management mobile app for patients with IA, by using a user-centred step-by-step approach and assess its real-life use.

**Data Availability Statement:** All relevant data are within the paper and its Supporting Information files.

**Funding:** This study was promoted and funded by the French Society of Rheumatology. The French Society of Rheumatology received an institutional grant from (in alphabetical order): Abbvie, Biogen, Janssen, Lilly, Merck Sharp & Dohme, Novartis, Nordic Pharma, Roche France and Pfizer. The pharmaceutical companies were not involved in the app's general design, development or content, study process, data collection, interpretation, manuscript writing or decision to publish. The grant was used to pay the researchers for the qualitative studies, the app development and communication. All other authors did not receive any honoraria for this study.

**Competing interests:** CB, TP, AL, CV, GD, PB, SD, FT, DL, LG, FAL, MA, SP and JS declare no competing interests in relation is this study. SG, FM were paid by Unknowns Research department for their contribution to the study. GM, JLC, MQ, were employed by Unknowns Conseil en stratégie et innovation, Paris, France HJ and LB are owners of Unknowns Conseil en stratégie et innovation, Paris, France Disclosures of interest apart from the study Catherine Beauvais reports research grants from BMS, Fresenius Kabi, Lilly, Mylan and was an occasional speaker for BMS, Abbvie, MSD, Mylan, Pfizer, Roche, Sanofi, UCB. She participated to a medical board for Sandoz and Novartis. Thao Pham reports speaker and consulting fees from Abbvie, Amgen, Biogen, BMS, Celgene, Fresenius-Kabi, Janssen, Lilly, MSD, Nordic, Novartis, Pfizer, Sandoz, Sanofi and UCB. Florence TUBACH is head of the Centre de Pharmacoépidémiologie (Cephepi) of the Assistance Publique – Hôpitaux de Paris and the Clinical Research Unit of Pitié-Salpêtrière hospital; both of these structures have received unrestricted research funding and grants for the research projects handled and fees for consultant activities from a large number of pharmaceutical companies that have contributed indiscriminately to the salaries of its employees. Florence Tubach is not employed by these structures and did not receive any personal remuneration from these companies. Jérémie Sellam reports fees from MSD, Pfizer, Abbvie, Fresenius Kabi, BMS,Roche Chugai, Sandoz, Lilly, Gilead, Novartis, Janssen, grant research fromSchwa Medico, MSD and Roche.

## Methods

The app development included first a qualitative study with semi-guided audiotaped interviews of 21 patients to identify the impact of IA on daily life and patient treatments practices and an online cross-sectional survey of 344 patients to assess their health apps use in general and potential user needs. A multidisciplinary team developed the first version of the app via five face-to-face meetings. After app launch, a second qualitative study of 21 patients and a users' test of 13 patients and 3 rheumatologists led to the app's current version. The number of app installations, current users and comments were collected from the Google Play store and the Apple store.

## Results

The qualitative study revealed needs for counselling, patient–health professional partnership, and skills to cope with risk situations; 86.8% participants would be ready to use an app primarily on their rheumatologist's recommendation. Six functionalities were implemented: a safety checklist before treatment administration, aids in daily life situations based on the French academic recommendations, treatment reminders, global well-being self-assessment, periodic counselling messages, and a diary. The Hiboot app was installed 20,500 times from September 2017 to October 2020, with 4300 regular current users. Scores were 4.4/5 stars at Android and iOS stores.

## Conclusion

Hiboot is a free self-management app for patients with IA developed by a step-by-step process including patients and health professionals. Further evaluation of the Hiboot benefit is needed.

## Introduction

Mobile health applications (mhealth apps) have undergone significant development in recent years and are of increasing interest and usefulness to help patients manage their chronic inflammatory arthritis (IA) [1–3]. IA—that is, rheumatoid arthritis (RA) and spondyloarthritis (SpA) including psoriatic arthritis (PsA)—represents painful chronic conditions impairing quality of life and work capacities. Disease-modifying anti-rheumatic drugs (DMARDs) are used to control IA disease activity, reduce functional disability and improve prognosis. They include methotrexate, biologic agents such as tumor necrosis factor (TNF) alpha blockers, and Janus-kinase (JAK) inhibitors. They are increasing in availability (more than 15 DMARDs in France) and have a wide variety of targets and modes of administration [4–8].

Most mhealth apps for IA described in the literature focus on symptom tracking, including disease activity, pain and fatigue [9–12], as well as physical activity [13, 14], treatment reminders and exchange functionalities [15]. Systematic reviews of IA apps have shown that most apps have been designed without involving patients in their development and that healthcare providers have rarely contributed [9, 10, 15]. The European Alliance of Associations for Rheumatology (EULAR) recently issued "points to consider," emphasizing that participation of patients and health professionals (HPs) was essential in the content, development and evaluation of mhealth apps [16]. Another challenge in mhealth apps relates to real-life use because

very few apps found in the literature are still available at stores after their launch [9], which suggests potential discrepancy between users' interests and the app design or content. The number of current installations and app users is rarely reported, although usage represents a basic assessment of their impact [13].

Few apps for IA patients have targeted patient education [9, 10, 15]. Patient education, including e-education, is advocated to develop self-care and improve patients' autonomy over their own health and treatment [17]. Apps may be appropriate tools for self-management such as medications management, problem-solving, and care coordination [3] or to offer a holistic approach of self-management [18]. An app providing information to help patients be more aware of their medications in daily life has not been addressed in IA. Patients receiving DMARDs are at risk of adverse events, including excess risk of infection, noted more in RA [19–21] than SpA [22], in particular because of co-medication with glucocorticoids and/or high disease activity. Measures patients can take to decrease these risks include vaccinations [23], self-referral and DMARD interruption in some situations [24–29] such as surgery, dental care or pregnancy, which patients need to discuss with HPs. An app helping patients to adopt appropriate behaviours with risk situations [30, 31] could be of interest. It could also help maintain patients' skills after face-to-face patient education sessions because patient education has been found effective in the short term only [32].

In this context, we developed a self-management mhealth app called Hiboot for patients with IA treated with DMARDs. The development was designed to involve patients and HPs, including the app concept, preliminary studies to explore patients' needs and understand the overall aspects of their daily lives, and use a step-by-step approach by adjusting the app in line with users' feedback. Evaluation was planned by collecting the number of app installations, regular users and scores and comments at app stores.

## Materials and methods

### Design of the Hiboot app development

The development was promoted by the French Society of Rheumatology and managed by a steering committee of 3 rheumatologists (JS, TP, CB), a member of a patient association (ST), a methodologist (FT) and members of a digital company including 3 anthropology researchers involved in qualitative studies (SG, FM, GM) (**Fig 1**). The development steps included a mixed-method qualitative–quantitative study (step 1) to obtain the first version of the app (step 2) and the app launch (step 3). After the app launch, users' tests and a second qualitative study were performed (steps 4 and 5), which led to the current version (step 6).

**Step 1. Mixed-method qualitative–quantitative study.** In the first qualitative study (May-June 2016), patients were recruited by the 3 rheumatologists of the steering committee and enrolled on a voluntary basis from 2 public hospitals and 2 private practices in France. Before the beginning of the study, the purposeful sampling methodology was used to obtain a variety of predetermined patient profiles by age, sex, disease duration, socio-professional status, type of IA (RA, SpA, PsA), type of DMARD (methotrexate or biologics) and number of previous DMARDs used. Assessment of medication adherence was not an eligibility criterion, nor was previous use of an mhealth app. After the interviews had begun, the sample was completed with the progression of the data collection, according to the approach of the grounded theory model [33]. A standardized semi-structured interview schedule (Appendix 1 in **S1 File**) explored 3 main areas: (1) daily life with IA, (2) practices of pharmacological and non-pharmacological treatments, and (3) the patient's social and work relationships. Eligibility criteria were adults (aged ≥18 years) with a diagnosis of IA according to the rheumatologist's opinion [34–36] who received methotrexate and/or biologic DMARDs (bDMARDs), were followed as

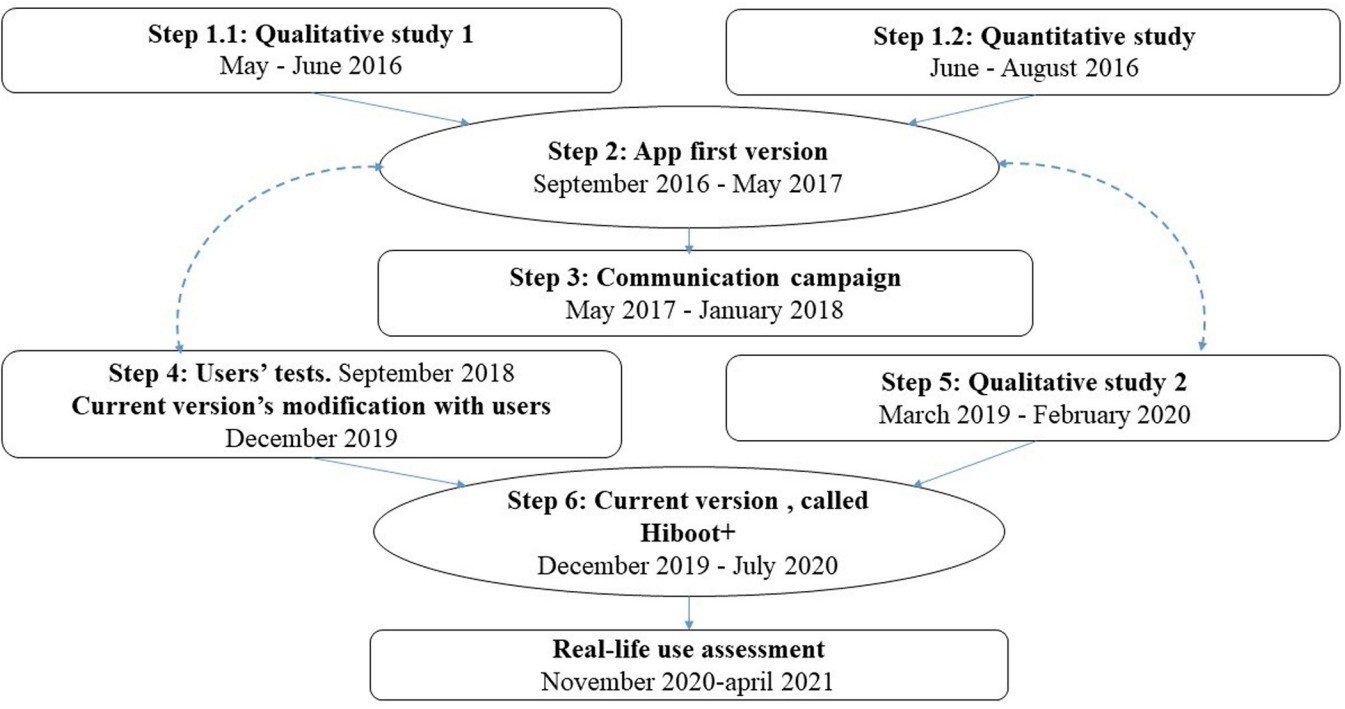

**Fig 1. App development and assessment timeline.**

outpatients or inpatients and were fluent in French. Exclusion criteria were conditions that could alter patients' understanding such as cognitive impairment and psychiatric disorders. Two anthropologists (GM, FM) conducted the interviews in the patient's usual environment (i.e. preferentially in their home). Interviews were audiotaped and transcribed verbatim. All data were de-identified to ensure confidentiality. Transcripts were analysed by using an ethnographic approach based on the grounded theory [33]. Patient enrolment was stopped at data saturation (i.e., when adding more participants did not provide any new information).

Data analysis and the consolidated criteria for reporting qualitative research criteria (COREQ) [37] are in Appendix 2 in **S1 File**.

The quantitative study (June-August 2016) was an online cross-sectional survey. The questionnaire was developed by the steering committee based on the study purpose. The survey content was drawn from the qualitative study with a focus on patients' practices with their treatments and the use of apps.

The broad areas of the survey were demographics, difficulties and problems encountered in daily life, needs for advice, participants' smartphone and mhealth apps use and motivation for an mhealth app for their IA, the app's potential content, and non-exhaustive issues in daily life. Inclusion criteria were a diagnosis of IA and treatment with DMARDs. The survey was posted on the 3 patient-association partner websites and Facebook accounts or were proposed anonymously by the investigating rheumatologists belonging to the 2 investigating hospital rheumatology departments and the 2 private offices, who provided the Internet link to patients. To check the inclusion criteria, patients were invited to state the name of their DMARDs. At analysis, the questionnaires with no available DMARD names were deleted. The quantitative descriptive analysis was performed with R Core Team (2017) (R: A language and environment for statistical computing. R Foundation for Statistical Computing, Vienna, Austria).

Associations between participants' willingness to use an mhealth app (grouped in 2 classes: "totally and rather agree" vs "rather not and not at all agree") and participants' characteristics were calculated by using Chi-square test or Fisher's exact test for categorical variables and Wilcoxon rank test for quantitative variables according to the statistical distribution.

**Step 2. Development of the first version of the app by a multidisciplinary team (September 2016-May 2017).** The team (including the steering committee) consisted of 7 rheumatologists, 3 patient association representatives and 5 members of the digital company and worked in 5 face-to-face meetings. The team discussed the implications of the preliminary studies (i.e. step 1) for the app development, and patient feedback helped define the look, features and navigation.

**Step 3. Launch of the first version of the app (May 2017- January 2018).** The launch involved a multimodal strategy including communication to patients, rheumatologists and health professionals. The first version of the app was released in May 2017.

**Step 4. Users' tests and development of the current version.** Two rounds of users' tests were conducted. The first round in September 2018 consisted of in-depth interviews of 7 patients of different profiles (current users or those who dropped out) and 3 rheumatologists, who recommended or not the app to their patients. The participants were recruited in the same way as in step 1. The second round of users' tests was conducted in December 2019, including 6 patients, 4 recruited by the rheumatologists and 2 recruited by Stephenson, an agency specialized in market analysis. The aim was to collect the patients' opinions on the app's features, interface and navigation and to investigate rheumatologists' opinions. Patient interviews took place in a neutral environment (café, workplace) or by phone. Rheumatologist interviews were by phone. The interviews were conducted with a predefined schedule. The main topics of the patient interviews were their comments on the use and functionalities of the app. The main topics of the rheumatologist interviews were their own knowledge of the app, their own feedback on the app and patient feedback, and barriers to recommending the app.

Step 5 was a second qualitative study, including RA patients receiving all types of DMARDs, added 2 years after the app's launch (March 2019- February 2020) to better understand patient needs following the availability in France of new DMARDs. Some of these new DMARDs, the targeted synthetic DMARDs (tsDMARDs), involved daily administration that could require modifying the app features. The schedule was built after a literature review of new issues related to tsDMARDs [38, 39]. The addressed domains were history of the patient pathway, the patient–doctor relationship, decision-making, and medication daily management (Appendix 3 in **S1 File**). Recruitment sampling, data collection and data analysis were conducted using the same methods as in Step 1.

**Step 6. Current version delivery.** The current version, called Hiboot+, was constructed during and/or after steps 4 and 5 with the inclusion of additional features and content, and released in December 2019.

## App assessment

The app's use was assessed by collecting data from app stores: Google play store for Android operating systems (https://play.google.com/console/) and Apple app store for iOS systems (https://appstoreconnect.apple.com/login). Both stores provide analytics features to monitor the performances of apps. The collected data included the number of installations from October 2017 to September 2020, defined as monthly installations minus monthly uninstallations; the number of regular users, defined as people whose app was in use at least once a month; the number and type of features used from June to November 2020; and the type of requests for

daily situations sought via the app from March to November 2020. Scores (0 to 5-star rating) were collected at the Google Play and Apple stores in January 2021. A content analysis of comments on the stores was conducted from launch to April 2021 and classified according to satisfaction/non-satisfaction, difficulties and missing functionalities.

# Results

## Qualitative studies

The first qualitative study enrolled 21 patients. Although the second qualitative study, also involving 21 patients, was conducted after the app launch, the emerging themes were similar. Therefore, results are presented together to avoid redundancy. Patients were mainly female (33/42), with median age 51 years (17–82): 33 had RA, 7 SpA and 2 PsA. Eleven received methotrexate monotherapy and 31 bDMARDs or tsDMARDs monotherapy or combined therapy with methotrexate (demographics in Appendix 4 in **S1 File**).

## Summary of the key emerging themes

Short quotes are available in Text box 1 and full quotes are in Appendix 5 in **S1 File**. The thematic analysis revealed 3 themes: (1) living with IA as a career, (2) acquisition of skills and lay knowledge and (3) patient treatment appropriation practices. Sub-themes (in italics) were identified and deduced for the app, each sub-theme corresponding to one or more of the app's functionalities (**Text box 1**).

Text box 1. Implications of qualitative studies for the app development

| Key themes | Sub themes | Short quotes | App functionalities |
|---|---|---|---|
| Living with IA as a career | *Search for causes. Looking for information* | *"When I was diagnosed I read everything, I wanted to know everything"* | Periodic messages on disease and treatment[1,2] |
| | | | Development by HPs[1,2] |
| | | | Relevant scientific sources [1,2] |
| | | | Promotion by the French Society of Rheumatology [1,2] |
| | *Patient–rheumatologist partnership* | *"We are partners: we have to fight and to stop the disease."* | Self-assessment[1,2] |
| | | | Comments on daily life events, disease activity and notes to be communicated to the rheumatologist[2] |
| | *Routines and habits* | *"Now I take it [methotrexate] on Saturday evening because it bothers me less."* | Treatment reminders[1,2] |
| | | | Self-assessment[1,2] |
| | | | Comments, medical appointments[1,2] |
| | | | Situational helps in daily life[1,2] |
| | | | Diary[2] |
| | *Side effects/disease complications. To handle situations in daily life. To know appropriate emergency responses* | *"I got a huge nail infection and had emergency surgery because I let it grow."* | Safety checklist before treatment administration[1,2] |
| | | | Situational aids in daily life[1,2] |

| | | | |
|---|---|---|---|
| Acquisition of skills and lay knowledge | *To navigate in the healthcare system* | *"I was in the middle of a crisis [. . .] and you can't get into the system"* | Periodic messages on disease and treatment[1,2] |
| | | | Situational aids in daily life[1,2] |
| | *To deal with information sources* | *"It's a marker for the disease [spondylitis] plus very high C-reactive protein"* | Development by HPs[1,2] |
| | | | Periodic messages on disease and treatment[1,2] |
| | | | Situational aids in daily life[1,2] |
| | | | Relevant scientific sources[1,2] |
| | | | Updated and reliable information[1,2] |
| | | | Understandable and simple messages[1,2] |
| | *To know how to collaborate with the rheumatologist and HPs* | *"I have learned to talk to my doctors"* | Self-assessment[1,2] |
| | | | Comments on daily life events, disease activity and notes to be communicated to the rheumatologist[2] |
| | | | Diary[2] |
| | *Experiments with the disease and treatment* | *"She [my rheumatologist] doesn't like it. But I do my own thing."* | Self-assessment[1,2] |
| | | | Treatment reminders[1,2] |
| | | | Comments on daily life events, disease activity and notes to be communicated to the rheumatologist[2] Diary[2] |
| | *Deal with the disease and complications, know how to cope with ordinary infections* | *"I have the maturity to recognize that. . . at the same time I can self-medicate"* | Safety checklist before treatment administration [1,2] |
| | | | Situational aids in daily life[1,2] |
| | | | Comments on daily life events, disease activity and notes to be communicated to the rheumatologist[2] |
| Patient treatment appropriation practices | *The treatment ritual* | *"I take a little 10 minute ritual even if the injection goes very fast"* | Treatment reminders[1,2] |
| | | | Diary[2] |
| | *Potential conflict between safety and adherence* | *"I hit my tibia with a chainsaw. . . I did my injection however, so, I was lucky. . . Etanercept has come into my life and it's a habit. . ."* | Safety checklist before treatment administration[1,2] |
| | | | Situational aids in daily life[1,2] |
| | *Influence of the daily mode of administration* | *"I might gain 1 or 2 years when I will be fine"* | Treatment reminders[1,2] |

IA, Inflammatory arthritis; HPs: health professionals.

[1]In first version of the app (after qualitative study 1).

[2]In the current version of the app (after qualitative study 2).

**1. Living with IA as a "career".** At the onset of the disease, participants reported a "career" starting with the *search for causes*, *looking for information* and *looking for the right diagnosis and the right doctor*.

IA management needed a close joint effort and a *patient–rheumatologist partnership*. Most patients developed *routines and habits* with their treatment and could face secondary lack of efficacy. The onset of *side effects or disease complications* could occur at any point in the disease course.

Patients had to cope with the unpredictability of these events and *to handle situations in daily life* such as infections or planned surgery. Patients needed to know what the *appropriate emergency responses* were and what situations required the rheumatologist's intervention or advice.

**2. Patient skills and lay knowledge.** To manage the situations at risk throughout their career, patients needed to develop various types of skills, some of which were related, as follows. *To navigate the healthcare system*: know which healthcare professional to consult, when and how often; *to deal with information sources*: know where to look for information, how to identify the most relevant sources; *to know how to collaborate with the rheumatologist* and other HPs: know what to expect and ask from them; *experiments with the disease and treatment*: patients were taking control over their medications via various adjustments and experiences such as changing their DMARD dosage or administration intervals on their own initiative or trying alternative medicine or food exclusions; and adaptation skills: *to deal with the disease and complications*, know *how to cope with ordinary infections* and other situations, manage minor treatment.

**3. Patient treatment appropriation practices.** Three additional sub-themes emerged:

*The treatment ritual*: patients ritualized their subcutaneous administration. Such rituals represented a reflexive moment, only dedicated to themselves and their disease:

*Potential conflict between safety and adherence*. Rituals tended to improve adherence but could lead to lack of concern about safety recommendations because of habits, over-confidence, and neglect of situations at risk.

*Influence of the mode of administration*. The daily administration of tsDMARDs (JAK inhibitor) did not modify patients' perception as compared with bDMARDs, but patients' opinions depended on their experience with RA (severity, activity, control).

## Implications for the app development

Six functionalities were implemented in the app: 5 from the first qualitative study: a safety checklist before treatment administration, aids to self-management in daily life or with risk situations, treatment reminders, global well-being self-assessment, and periodic counselling messages; and an additional one from the second qualitative study: the diary was added to note comments, appointments and other treatments.

## Quantitative study

The quantitative study included 344 patients, and 331 questionnaires were complete for analysis: 82.8% of patients were female, 55.7% had RA, and 62.6% received methotrexate and 70.4% bDMARDs (**Table 1**); 238 (78%) patients had a smartphone and 191 (80.9%) were using apps, but only 61 (32.3%; 18.4% overall) were using mhealth apps (**Table 2**). Whether or not they had used apps, 70.5% of the patients reported questions or difficulties with their treatment and 67.5% had needed help or advice. The main issues were infections, vaccines and surgery, whereas storage/travel, dental care or missed doses were rarely reported. Fatigue and the wish to stop treatment were themes related to patients' way of coping with their disease.

Among app users who answered the question (n = 211), 86.8% (n = 177) would willingly use an app to manage their treatment (51% strongly agreed, 35.8% rather agreed) and 64.4% (n = 112) would accept the app only on their rheumatologist's recommendation (**Table 3**). Participants interested in the app were more frequently younger than those who would not use the app [age 48.4 years (± 13 SD) vs 54.0 years (± 12 SD), Wilcoxon rank sum test $p < 0.05$], not members of a patient association (66.7% vs 45.4%, chi-square test $p < 0.05$) and lived in medium-sized than other sized cities (Fisher exact test $p < 0.01$). We found no association with other socio-demographic characteristics, level of education, type/duration of arthritis or knowledge.

**Table 1. Quantitative study.** Patient characteristics: demographics, clinical features and other information (n = 344). The quantitative study was carried out from June to August 2016.

| | Total respondents | Count |
|---|---|---|
| Female | 331 | 274 (82.8) |
| Age (years), mean ± SD | 317 | 53.49 ± 13.8 |
| Professional activity | 323 | |
| Currently employed | | 130 (40.2) |
| Retired | | 103 (31.9) |
| On sick leave/disability | | 90 (27.9) |
| Socio-professional status | 326 | |
| Higher | | 93 (28.5) |
| Lower or intermediate | | 221 (67.8) |
| Other | | 12 (3.7) |
| Size of place of residence (No. inhabitants) | 321 | |
| ≥ 200 000 | | 63 (19.6) |
| 10 000–199 999 | | 123 (38.3) |
| < 10 000 | | 135 (42.0) |
| Education level | 320 | |
| High school or less | | 119 (37.2) |
| University | | 201 (62.8) |
| Member of a patient association (yes) | 331 | 197 (59.5) |
| Information sources about disease or treatments: | 344 | |
| General practitioner | | 94 (27.3) |
| Rheumatologist | | 344 (100) |
| Face-to face/group patient education including nurses | | 157 (45.6) |
| Other healthcare practitioner | | 75 (21.8) |
| Internet/media/ papers | | 236 (68.6) |
| Brochures or leaflets/books | | 344 (100) |
| Type of disease, n (%) | 298 | |
| Rheumatoid arthritis | | 166 (55.7) |
| Axial or peripheral spondyloarthritis (including psoriatic arthritis) | | 119 (39.9) |
| Other | | 13 (4.4) |
| Disease duration (years), mean ± SD | 291 | 13.81 (12.49) |
| *Treatments* | 324 | |
| Current methotrexate | | 203 (62.6) |
| Current methotrexate duration (years), mean ± SD | 190 | 8.79 (8.09) |
| Oral | | 66 (32.5) |
| Subcutaneous | | 131 (64.5) |
| Injection by the patient | | 54 (41.2) |
| Do you sometimes forget to take your methotrexate? (yes) | 181 | 41 (22.6) |
| Current bDMARD | | 228 (70.4) |
| Current bDMARD duration (years), mean ± SD | 222 | 6.91 (5.79) |
| Subcutaneous bDMARD | | 169 (74.1) |
| Injection by the patient (yes) | | 123 (72.7) |
| Do you sometimes forget to take your bDMARD?[3] (yes) | 199 | 38 (19.1) |
| Intravenous bDMARD | | 57 (25.2) |
| Both methotrexate and bDMARDs | | 111 (34.3) |
| Disease activity self-assessment (NRS[1] 0–10) ˣ, mean ± SD | 331 | 4.43 (2.44) |

*(Continued)*

**Table 1.** (Continued)

| | Total respondents | Count |
|---|---|---|
| Coping[2] (NRS[1], 0–10) ʸ, mean ± SD | 328 | 3.83 (2.33) |

Data are n (%) unless otherwise indicated.

[1]NRS, numeric rating scale. [2] Coping derived from the RAID score [1]. [3]Among patients receiving subcutaneous bDMARDs. ʸ High score means high disease activity or bad coping.

[1] Gossec L, Paternotte S, Aanerud GJ, Balanescu A, Boumpas DT, Carmona L, et al. Finalisation and validation of the rheumatoid arthritis impact of disease score, a patient-derived composite measure of impact of rheumatoid arthritis: a EULAR initiative. Ann Rheum Dis. 2011;70:935–42

The app would be used (**Table 3**) to find out what to do in case of risk situations (92.8%, n = 196 [strongly or rather agreed]) or to find out what symptoms would require stopping treatment (82%; n = 173). It could also be used as a treatment reminder (79.7%; n = 162), a reminder of the monitoring tests (89.9%; n = 184) or as a safety checklist before treatment administration (77%, n = 156).

## Development of the first version of Hiboot

The *development* of the app (Step 2) is presented in **Text box 2**.

## Text box 2. Hiboot functionalities and content

| Functionalities and content | First version | Current version |
|---|---|---|
| Available treatments | Methotrexate, TNF blockers (originators and biosimilars), tocilizumab, abatacept, rituximab (originators and biosimilars) | Methotrexate, TNF blockers (originators and biosimilars), tocilizumab, abatacept, rituximab (originators and biosimilars), tofacitinib, baricitinib, ustekinimab, secukinumab |
| Home page | • | •• |
| Checklist before administering treatment | • | •• |
| Aids in daily and risk situations | • | •• |
| Educational messages | • | •• |
| Treatment reminder | • | •• |
| Other treatment monitoring | | •• |
| Diary: medical appointments | | •• |
| Diary: blood test appointments | | •• |
| Diary: other kinds of appointments | | •• |
| Diary: treatment orders | | •• |
| Diary: personal comments, events, memos | | •• |

TNF, tumor necrosis factor • availability in the first version. •• modified or added in the current version.

*Information on COVID-19 could not be pursued on Android after June 2020 because of cancellation by Google.

**Table 2. Quantitative study: Smartphone use, difficulties encountered and information needs in daily life (n = 344).**

| | Total respondents | Count |
|---|---|---|
| Do you have a SmartPhone (iPhone or Android or Windows phone)? [1] (yes) | 305 | 238 (78.0) |
| Do you generally use applications on your smartphone?[2] (yes) | 236 | 191 (80.9) |
| Frequency of application use | 166 | |
| Once a day | | 47 (28.3) |
| 2 to 10 times a day | | 86 (51.8) |
| > 10 times a day | | 33 (19.9) |
| Do you use health-related applications? [3] (yes) | 189 | 61 (32.3) |
| Have you experienced any difficulties or any questions about your treatment? [1] (yes) | 312 | 220 (70.5) |
| Facing these difficulties or questions, did you feel the need for advice or counselling? [1] | 280 | 189 (67.5) |
| Mark below the situation(s) that you have encountered or you have had problems with: [4] | 312 | |
| Fever or infection | | 156 (25.7)[5] |
| Vaccines | | 76 (12.5) |
| Surgery | | 59 (9.7) |
| Storage/travelling | | 50 (8.2) |
| Fatigue | | 48 (7.9) |
| Wish to stop treatment | | 45 (7.4) |
| Dental care | | 43 (7.1) |
| Drug administration | | 34 (5.6) |
| Forgotten dose or missed dose | | 29 (4.8) |
| Gastrointestinal symptoms | | 30 (4.9) |
| Pulmonary symptoms | | 19 (3.1) |
| Pregnancy planning | | 12 (2.0) |
| Other symptoms and adverse effects | | 6 (1) |
| What other information would be useful to you? Indicate without restriction your opinion on other possible uses of the application.[2] | | 30[6] |

Data are n (%) unless otherwise indicated.

[1] Among the total population.

[2] Among individuals with a smartphone.

[3] Among individuals using applications.

[4] Among individuals facing difficulties.

[5] In total, 607 situations were collected. The % is calculated relative to the total number of situations.

[6] Self-monitoring (pain, difficulties, drug monitoring) (10 items), management of adverse effects (8), holistic management, improve daily life (6), new treatments (3), patient–doctor communication (2), chat box (2), adherence (1).

**Type of application.** Considering legal issues, the steering committee and the French Society of Rheumatology decided that no personal information would be collected. Consequently, the app was a self-administered app for self-management and was not registered as a medical device. Patients' data were not to be directly communicated to rheumatologists or general practitioners, but patients could make screenshots of their app to communicate with their physician during consultations. The app was free of charge. Terms and conditions of use informed the patients of the app type and that information content intended to help in daily situations but did not provide medical advice or consultation; did not constitute personal advice, diagnosis, aid to diagnosis, treatment or aid to treatment; and did not constitute or

**Table 3. Quantitative study: Potential use of the app, patients' approval[1], and implications for app development[2] (*in italics*) (n = 236).**

| Potential use | Total respondents | Totally agree | Rather agree | Rather not agree | Not at all agree |
|---|---|---|---|---|---|
| I would willingly use this application | 204 | 104 (51.0) | 73 (35.8) | 13 (6.4) | 14 (6.9) |
| *I would only accept the app on the recommendation of my rheumatologist* | 211 | 49 (28.2) | 63 (36.2) | 27 (15.5) | 35 (20.1) |
| I would use it: | | | | | |
| *In case I had symptoms that would require stopping my treatment* | 211 | 101 (47.9) | 72 (34.1) | 22 (10.4) | 16 (7.6) |
| *To know the situations at risk with my treatment* | 211 | 113 (53.5) | 83 (39.3) | 6 (2.8) | 9 (4.3) |
| *To have a treatment reminder* | 202 | 106 (52.5) | 56 (27.2) | 23 (11.4) | 17 (8.2) |
| *To know what to do in case of missed doses or forgotten treatment* | 199 | 86 (43.2) | 69 (34.7) | 27 (13.6) | 17 (8.5) |
| *To have a safety checklist before treatment administration* | 201 | 82 (40.8) | 74 (36.2) | 27 (13.3) | 18 (9.0) |
| *To recall how to make self-injections* [3] | 176 | 61 (34.6) | 56 (31.8) | 22 (12.5) | 37 (21.0) |
| *To have a reminder on monitoring tests* | 205 | 104 (50.7) | 80 (39.2) | 8 (3.9) | 13 (6.3) |
| *To get information on reliable websites* | 211 | 101 (47.9) | 80 (37.9) | 18 (8.5) | 12 (5.7) |

Data are n (%).

[1] Among individuals with a smartphone.

[2] Implications for app development are in Text box 1.

[3] not yet implemented in the app

substitute for advice, diagnosis or recommendations provided by a competent HP. The promotor (i.e., French Society of Rheumatology) was mentioned.

## Hiboot features and content

The choice was for a "companion" called Hiboot (Owl), whose interface should not appear too medical. The owl evokes a watchful animal and because of its small size, it could be a companion (Hiboot screenshots in **Fig 2** and in Appendix 1 in **S2 File**). The potential use by patients with hand deformity was addressed during the conception.

*Safety checklist before treatment administration* intended to warn patients about situations that could contraindicate treatment intake (Appendix 2 in **S2 File**). Six types of risk situations were identified. If one situation was quoted, an explanation of the situation was provided and the patient was invited to not take the treatment and to contact his/her doctor.

*Aid in daily life situations including risk situations* (examples in Appendix 3 in **S2 File**). Situations requiring patient information were identified from the quantitative study and from a previous work on patients' safety skills by the patient education study group of the French Society of Rheumatology [40, 41]. They were incorporated into the app by keywords commonly used by the public (Appendix 3 in **S2 File**). The aids were aimed as warnings to patients to promote self-referral to HPs: information included a general presentation of the situation and advice on what to do and when/how to refer to the rheumatologist, hospital team or the family doctor. A "patient phrasing" was built by consensus to be understandable by patients and the public.

For each situation, 3 information materials were selected: summary of product characteristics, material validated by national health authorities [42–45], and guidelines of the French Society of Rheumatology and the *Club Rhumatismes et Inflammations* (CRI). [24–28, 46]. DMARDs and IA treatments that did not have guidelines from the CRI could not be implemented in the app. Therefore, the app content was free from commercial, unverified or non-consensual sources among rheumatologists. Details on the app's information sources are in Appendix 4 in **S2 File**.

*Educational messages* (examples in Appendix 5 in **S2 File**) consisted of general information on the disease, healthy lifestyle, daily life matters and symptomatic treatments of IA. The topics were

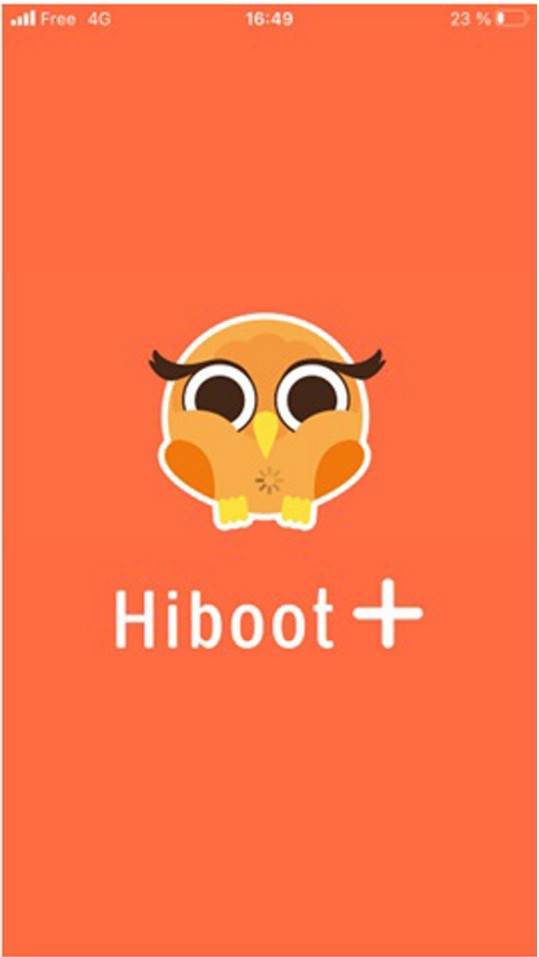

**Fig 2. HIBOOT home page.** Republished from a screenshot of the application home page under a CC BY license, with permission from the French society of rheumatology original copyright 2022.

extracted from the websites of the partner patient organisations and/or directly from the patients/ participants in the team and from previous work on essential knowledge for patients with IA [47].

*Treatment reminders* were used to remind about the app, medications use and the safety checklist. The app users had to input their treatment to receive the reminders.

*Self-assessment* involved a simple question, "How do you feel today?", which was asked at the time of the reminder and checklist and answered on a 5-point Likert scale, from 1, badly, to 5, perfectly well. The answers were stored only in the app.

The compete files of the checklist, situational aids and educational messages are available upon request.

## Implementation technology

The Hiboot app is a hybrid app. Development was made with the javascript front-end framework AngularJS, then packaged by using Cordova. We used an Ionic library to create the user interface. The Ionic library runs on a wide range of Android and iPhone devices. Content of the app was managed by a headless content management system named Prismic. User data and parameters are stored within the app.

### The launch campaign

The launch campaign (Step 3) included emails to the rheumatologists on the French Society of Rheumatology mailing list, monthly newsletters from the CRI network, printed flyers and posters for outpatient clinics in private rheumatology offices and departments. Advertising was produced for the front page of the French Society of Rheumatology website and for social networks (Facebook, Twitter @AppliHiboot) as well as on patient association websites and Facebook/Twitter accounts. A video was posted on YouTube. The first version was launched in May 2017, with small changes (bug corrections) made until January 2018 (Fig 1).

### Users' tests

Demographics of participants in the users' tests and quotes (Step 4) are in Appendices 1 to 4 in **S3 File**. We interviewed 13 patients (7 using the app, 2 who dropped out and 4 non-users) and 3 rheumatologists. Patient interviews were from 90 to 120 min and rheumatologist interviews from 30 min to 1 hr.

The tests showed 2 usage types: (1) regular use by patients who were regularly using the app and asked for more features and (2) occasional use for specific questions requiring quick answers. The features valued by participants were the reminder system, the situational aids and the messages. However, some patients found the checklist and treatment recalls too repetitive and preferred optional use. The navigation system also had to be improved. Some patients also wanted more holistic self-management by using a personal diary.

Rheumatologists found it difficult to identify the scope of the app to recommend it routinely because it requires a proactive attitude and personal experience of its content.

### Current version

The current version (Step 6, **Text box 2**) was launched in December 2019 on iOs and January 2020 on Android, with bug corrections until July 2020 (Fig 1). It implemented additional DMARDs including anti-IL-17 blockers, JAK inhibitors and biosimilars, updated information on all DMARDs, and added functionalities such as a diary to note medical appointments, other personal medications, comments, life events and memos to prepare for medical consultations. A website was developed [48] to present the situational aids, the other functionalities remaining only available on the app.

### Assessment of the use of the app

The app was installed 20,500 times from October 2017 to September 2020 from both stores (Android and iOS) with an increasing curve (**Fig 3**). The number of app removals was 2870. The number of regular users was 4328 (i.e. approximately 21% of the installations).

From March to November 2020, users made 18,000 requests about daily life situations (**Table 4**). Most requests corresponded to those identified in preliminary studies, such as infections, dental care, and storage/travel. However, the first troublesome symptoms searched in the app were skin allergies, injection site reactions and tattoos and piercing, which had not been identified before development. Questions on libido ranked in sixth position. This topic had been added after users' feedback, as had thermal/spa therapy. In March 2020, a peak in use concerned COVID-19. Information on COVID-19 could not be pursued on Android after June 2020 because of cancellation by Google with the claim that Hiboot was not a government app.

The most-used functionalities were the checklist (3.97 views/month/user), diary (3.18 views/month/user) and daily/at risk situations (3.38 views/month/user) (**S4 File**). The scores

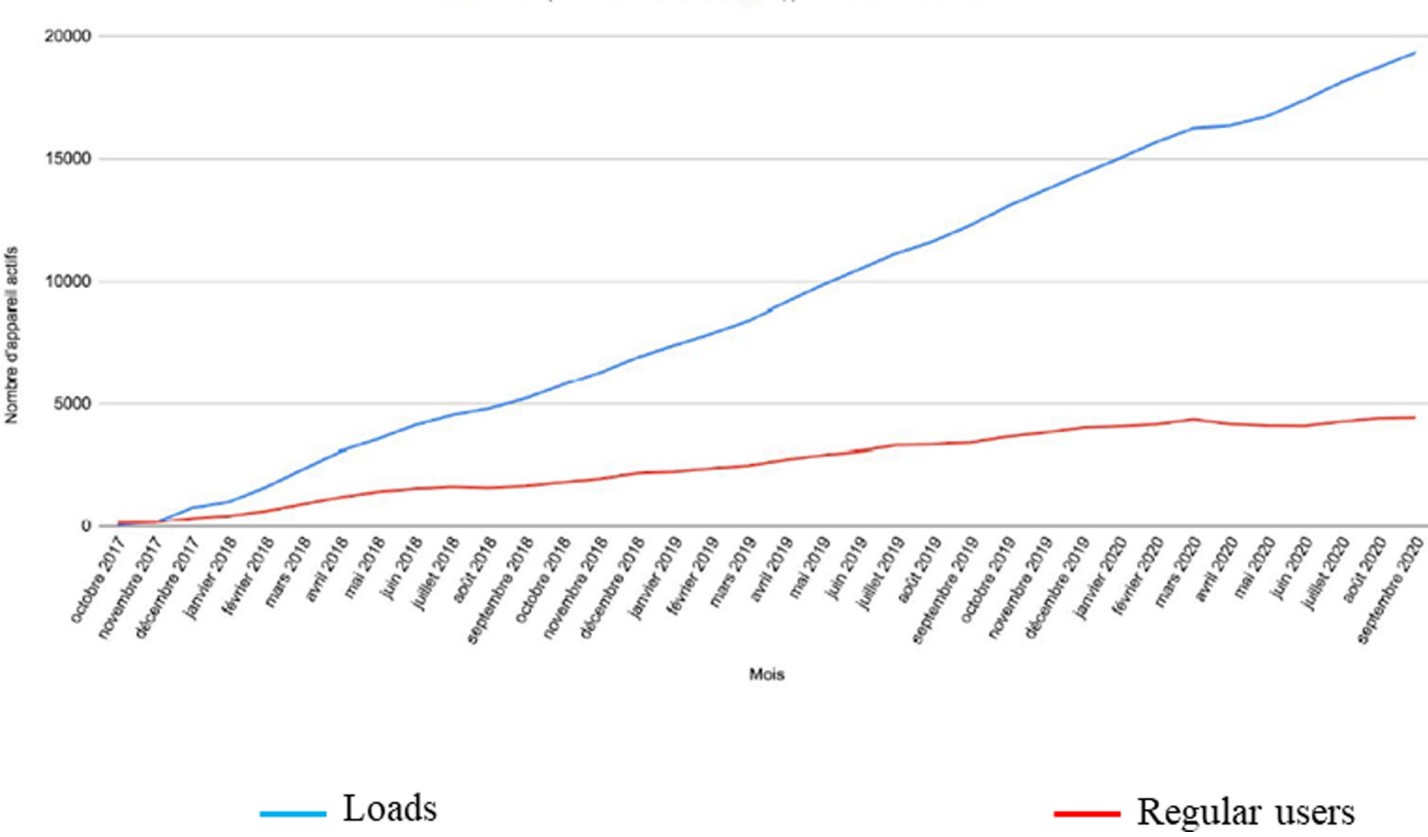

**Fig 3. Number of installations and mean number of regular users* from October 2017 to September 2020.** * Regular user: individual using the app once a month or more (data from Android + iOS).

at both stores have been 4.4/5, with most 4 and 5 (Appendix 1 in **S5 File**). The analysis of the 124 comments on the app stores (Appendix 2 in **S5 File**) showed 47 overall positive comments with particular "likes" for the app initiative, the reminders, the checklist and advice. The 8 overall negative comments were mainly related to the bugs in the app updates. Fifty-five bugs were reported or complained about: 19 bugs were reported on the reminder system, particularly on Android where 15 bugs owed to the battery optimisation function, which could block reminders and 10 bugs were reported about the checklist in iOS, which had a presentation defect. These bugs were corrected. Fourteen users regretted missing features such their treatment not included in the app. A few users wanted to be able to print/save the diary content.

## Discussion

Here we report the development of the Hiboot app dedicated to self-management for patients with IA. The design and process followed a step-by-step strategy matched with patients' opinions by using 2 qualitative studies, one quantitative study and users' tests to improve versions over time. The final device contains 6 functionalities, including a safety checklist, aids for everyday life and risk situations, treatment reminders, simple self-assessment, educational messages, and a diary to note comments, appointments and other medications. Hiboot is a free app. Real-life use assessment showed a still-increasing number of current users, 4300 in September 2020, and good scores at app stores.

**Table 4. Information on daily situations/symptoms and patient queries from the app from March to November 2020.** Data collected in Google analytics.

| Situations/symptoms | First version | Current version | Key words in current version (no.) | General information[1] | Specific information[2] | Patients' queries (no.) | Ranking of queries |
|---|---|---|---|---|---|---|---|
| **Total situations/symptoms (no.)** | **19** | **34** | | | | **18022** | |
| Total key words (no.) | | | 271 | | | | |
| **Skin abnormalities** | | | **49** | | | **3263** | 1 |
| *Sensitivity to sunlight* | • | • | *12* | • | • | *1011* | |
| *Skin reactions, allergies* | • | • | *15* | • | • | *1304* | |
| *Burns* | • | • | *5* | • | • | *185* | |
| *Wounds* | • | • | *8* | • | • | *604* | |
| *Mycosis/candidiasis* | | • | *9* | • | • | *159* | |
| **Infections** | | | **64** | | | **3082** | 2 |
| *Influenza* | • | • | *11* | • | • | *540* | |
| *Bronchitis, pneumonia* | • | • | *12* | • | • | *446* | |
| *Gastroenteritis* | • | • | *9* | • | • | *379* | |
| *Urinary tract infections* | • | • | *12* | • | • | *328* | |
| *Herpes, chickenpox, shingles* | • | • | *7* | • | • | *130* | |
| *Vaccines* | • | • | *23* | • | • | *435* | |
| *COVID-19* | | • | *2* | • | | *824* | |
| **Dental care** | • | • | **16** | • | • | **2701** | 3 |
| **Questions about treatment** | | | **13** | | | **2129** | 4 |
| *Forgetting treatment* | • | • | *7* | • | • | *945* | |
| *Stopping treatment* | • | • | *2* | • | • | *566* | |
| *Drug Interactions* | • | • | *4* | • | • | *618* | |
| **Storage/travel** | | | **22** | | | **1769** | 5 |
| *Storage at home* | • | • | *5* | • | • | *451* | |
| *Carrying treatment* | • | • | *8* | • | • | *624* | |
| *Going on a journey* | • | • | *9* | • | • | *694* | |
| **Libido/erectile dysfunction** | | • | **8** | • | • | **1698** | 6 |
| **Surgery** | • | • | **15** | • | • | **805** | 7 |
| **Pregnancy** | • | • | **11** | • | • | **638** | 8 |
| **Thermal/spa therapy** | | • | **5** | • | | **599** | 9 |
| **Blood sample abnormalities** | | | **26** | | | **466** | 10 |
| *Blood abnormalities* | | • | *11* | • | • | *171* | |
| *Increase in cholesterol levels* | | • | *6* | • | • | *86* | |
| *Liver/hepatic abnormalities* | | • | *9* | • | • | *209* | |
| **Gastrointestinal symptoms** | | | **20** | | | **445** | 11 |
| *Digestive disorders* | | • | *10* | • | • | *211* | |
| *Blood in the stools* | | • | *5* | • | | *57* | |
| *Colonoscopy/fibroscopy* | | • | *5* | • | | *177* | |
| **Other symptoms** | | | **22** | | | **427** | 12 |
| *Fatigue* | | • | *1* | • | • | *147* | |
| *Fractures* | | • | *5* | • | • | *48* | |
| *Cancers* | | • | *8* | • | | *81* | |
| *Dizziness* | | • | *4* | • | | *110* | |
| *High blood pressure* | | • | *4* | • | • | *41* | |

[1] information common to all DMARDs.

[2] Information depending on the type of DMARD.

• availability

From the preliminary studies (step 1), features that needed to be included in the app were information seeking, the HP–patient relationship, and assistance in everyday situations. Maintenance of patient skills in risk situations could conflict with good adherence to DMARDs by habits and over-confidence. Therefore, medication reminders and patient self-assessment at the time of ritualizing DMARD self-administration could constitute additional incentives to use the safety checklist. However, the systematic use of the checklist was counterproductive for some patients as users' tests showed, so the checklist was secondarily made optional.

Preliminary studies showed that reliable information was important for patients, which was consistent with other studies [49, 50]. The app was then designed to be free from commercial or non-consensual sources among rheumatologists so that they could recommend the app. Besides self-management of potential risks, other features were found relevant to implement related to experiments currently practiced by people with IA (e.g., missed doses or the wish to stop or taper DMARDs). Patients' concerns represented a wide range of everyday situations, some of which emerged after the launch and included minor but embarrassing skin problems or more intimate sexual issues.

To our knowledge, no self-management app of this nature is currently available for adult patients with IA, particularly with respect to medication management. The design, process and evaluation of the app can be compared to those described by Cai et al. [18], who developed an app for young patients with juvenile idiopathic arthritis and followed a user-centred approach with qualitative in-depth studies.

The Hiboot app meets 6 of the 10 EULAR recommendations for mhealth app development for people with IA [16]: to date, scientifically justifiable, user acceptable and evidence-based information (recommendation 1); relevant and tailored to individual needs (2); involvement of patients and HPs in design and development (3); transparency of the app's developer, funding source, content validation process, version updates and data ownership (4); attention to data protection (5); and facilitation of patient–healthcare provider communication (6).

The app content covers many aspects of self-management, with a large part devoted to treatment self-management skills [30, 40, 41]. With reference to the well-established definition of self-management [51, 52] that is, "the ability of the individual to manage symptoms, treatment, lifestyle changes and psychosocial and cultural consequences of health conditions", Hiboot meets the objective of helping IA people with their treatments, symptoms and information needs and provides healthy lifestyle and daily life messages.

The Hiboot app does not monitor patient-related outcomes as described in other apps [12]. Self-assessment in the app is a simple question, "How do you feel today", to be eventually shared with the rheumatologist or physician. Studies have shown that patients favoured self-monitoring of health or disease activity [49, 50, 53] and many people with IA would agree to share their mobile app data for research purposes or regularly enter data [50]; however, only 10 patients in our survey suggested patient-related outcome self-monitoring, so it was not retained in the app. Regardless, no specific question was asked about this feature in the survey, which might have underestimated patients' interests.

Hiboot provides treatment recalls similar to the literature [15]. Including reminders was of interest because medication adherence in IA needs to be improved [54, 55] and the benefit of reminders has been shown by short message service systems in RA [56] or by apps for other chronic conditions [57].

The collection of the number of installations and regular users at stores is a strength because this was seldom described in the literature; other strengths are the collection of comments at stores, used functionalities and people's queries. The 20,500 installations may seem small as compared with the IA prevalence in France, estimated at 400,000 people for both RA and SpA [58, 59], so approximately 5% of patients installed the Hiboot app. However, this number

seems substantial as compared with other apps for people with IA dedicated to physical activity: among 3 apps identified by Bearne et al. [13], one was installed by 1000 people with RA and one by 500 people; information was not available for the last one. The number of installations is also to be compared with the still low use of mhealth apps: 4.1% of the IA population used mhealth apps in Germany [50] and 4.6% of RA patients in France [60]. This observation may indicate that the number of people who installed the Hiboot app was close to the whole population who already use mhealth apps (i.e, 5%). The app's life is longer than most published apps: some have shown a rapid decrease in use over a 4-week period [12]; 75% people had stopped using them after 3 months [49] and only 5% of apps for rheumatic and musculoskeletal diseases were still available 2 years after their launch [9].

With regard to EULAR recommendations, 4 recommendations have not been met so far. Hiboot is not a medical device collecting data for research or self-monitoring of disease activity. As mentioned above, this aspect did not emerge from our preliminary studies but may have been underestimated. Although comments at stores did not reveal negative comments (apart from reports or complaints about "bugs"), a further study will be necessary to assess the benefit/risks of the app's use (EULAR recommendation 6) and the cost–benefit balance (recommendation 10). The design of the app could not meet recommendation 9 on providing a social network, which is only an optional recommendation. Accessibility of people across ages and abilities (recommendation 8) is a challenge in developing apps: this has been well quoted by rheumatologists about the difficulty in recommending the app. Not checking health literacy was also a limitation, because another study mentioned this as an important barrier to the use of apps [50].

For users' assessment, we looked for only star ratings and comments at stores. We are aware that this is very broad information that poorly reflects the true app quality as compared with other evaluation tools, the Mobile Application Rating Scale (MARS) being the most commonly used [61, 62]. The star systems may over-rate the app quality for two thirds of users [63]. Although the MARS tool has a user version [62], collecting spontaneous and unselected users' comments appeared of interest. The comments showed that Hiboot was not spared from "bugs" and had to be revised, particularly on the Android platform. In all cases, future assessment of the benefit of the app as to its features, safety and adherence will be needed.

In conclusion, the Hiboot app is a free app dedicated to self-management for patients with IA. The app has good usage and was developed according to international recommendations for people with IA. Further studies will be needed to assess the benefits of this app.

## Declarations

**Ethics and consent to participate.**   Participants of the qualitative research were informed of the study objectives and schedule and expressed their non-opposition according to local recommendations. The research was approved by an ethics committee (*Comité de Protection des Personnes*, no. 19.07.02.68617) and was declared to the commission for data collection, the *Commission Informatique et Liberté* (no. 2,214,586). Regarding the quantitative survey, patients who were given the internet link by the rheumatologists were orally informed of the objectives of the study. Their participation was not monitored. Therefore, approval by an ethics committee was not requested according to local recommendations.

## Supporting information

**S1 File. Qualitative studies.**
(DOCX)

**S2 File. Hiboot features and content examples.**
(DOCX)

**S3 File. Users' tests.**
(DOCX)

**S4 File. Number of views/users\*/months from June to November 2020.**
(DOCX)

**S5 File. Scoring and comments at app stores.**
(DOCX)

## Acknowledgments

We thank all patients who participated the qualitative and quantitative studies; patients and rheumatologists who participated in the users' tests and app improvements; Mme Nathalie Robert from the association Spondyl(O)action for users' feedback; Mahed Iqbal (Unknowns) and Alexandre Galatioto (Unknowns) for development contribution; Damien Carnet (ANDAR) for posting the quantitative study questionnaire; Laura Smales for reviewing the English version; and Frédéric Lioté, Hôpital Lariboisière, Paris, who contributed to patient recruitment in the second qualitative study.

Membership of Therapeutic patient education group of the French Society of Rheumatology and Club Rhumatismes et Inflammations are available at www.cri-net.com and https://sfr.larhumatologie.fr/.

## Author Contributions

**Conceptualization:** Catherine Beauvais, Thao Pham, Alexandre Lafourcade, Henri Jeantet, Ludovic Besset, Jérémie Sellam.

**Data curation:** Catherine Beauvais, Thao Pham, Guillaume Montagu, Sophie Gleizes, Francesco Madrisotti, Alexandre Lafourcade, Jérémie Sellam.

**Formal analysis:** Catherine Beauvais, Thao Pham, Guillaume Montagu, Alexandre Lafourcade, Marie Antignac, Jérémie Sellam.

**Funding acquisition:** Catherine Beauvais, Thao Pham, Jérémie Sellam.

**Investigation:** Catherine Beauvais, Thao Pham, Guillaume Montagu, Sophie Gleizes, Francesco Madrisotti, Alexandre Lafourcade, Céline Vidal, Guillaume Dervin, Pauline Baudard, Sandra Desouches, Delphine Lafarge, Laurent Grange, Françoise Alliot-Launois, Sonia Tropé, Jérémie Sellam.

**Methodology:** Catherine Beauvais, Thao Pham, Guillaume Montagu, Florence Tubach, Henri Jeantet, Marie Antignac, Ludovic Besset, Jérémie Sellam.

**Project administration:** Catherine Beauvais, Thao Pham, Sonia Tropé, Jérémie Sellam.

**Resources:** Catherine Beauvais, Thao Pham, Jérémie Sellam.

**Software:** Florence Tubach, Julian Le Calvez, Marie de Quatrebarbes, Henri Jeantet, Ludovic Besset.

**Supervision:** Catherine Beauvais, Thao Pham, Guillaume Montagu, Ludovic Besset, Jérémie Sellam.

**Validation:** Catherine Beauvais, Thao Pham, Guillaume Montagu, Alexandre Lafourcade, Jérémie Sellam.

**Visualization:** Catherine Beauvais, Guillaume Montagu, Sophie Gleizes, Francesco Madrisotti, Alexandre Lafourcade, Jérémie Sellam.

**Writing – original draft:** Catherine Beauvais, Thao Pham, Guillaume Montagu, Alexandre Lafourcade, Jérémie Sellam.

**Writing – review & editing:** Catherine Beauvais, Thao Pham, Guillaume Montagu, Sophie Gleizes, Francesco Madrisotti, Alexandre Lafourcade, Céline Vidal, Guillaume Dervin, Pauline Baudard, Sandra Desouches, Florence Tubach, Julian Le Calvez, Marie de Quatrebarbes, Delphine Lafarge, Laurent Grange, Françoise Alliot-Launois, Henri Jeantet, Marie Antignac, Sonia Tropé, Ludovic Besset, Jérémie Sellam.

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
