## [Decision Letter · Decision Letter 0]

9 Dec 2021

PONE-D-21-29287Development and real-life use assessment of a self-management smartphone application for patients with inflammatory arthritis. A user-centred step-by-step approach.PLOS ONE

Dear Dr. Beauvais,

Thank you for submitting your manuscript to PLOS ONE. After careful consideration, we feel that it has merit but does not fully meet PLOS ONE’s publication criteria as it currently stands. Therefore, we invite you to submit a revised version of the manuscript that addresses the points raised during the review process. Both the reviewers are suggesting to report changes, especially in the methods section. Several suggestions were provided by R1, fewer suggestions by the second reviewer but still, some relevant indications are reported. Nevertheless, it seems to me that the reviewers agree that the work has merit. For these reasons, I encourage you to review the article following the indications of the reviewers and resubmit.

Regards

We look forward to receiving your revised manuscript.

Kind regards,

Simone Borsci, Ph.D.

Academic Editor

PLOS ONE

Journal Requirements:

Reviewers' comments:

Reviewer's Responses to Questions

**Comments to the Author**

1. Is the manuscript technically sound, and do the data support the conclusions?

Reviewer #1: Partly

Reviewer #2: Yes

2. Has the statistical analysis been performed appropriately and rigorously? 

Reviewer #1: Yes

Reviewer #2: Yes

3. Have the authors made all data underlying the findings in their manuscript fully available?

Reviewer #1: Yes

Reviewer #2: Yes

4. Is the manuscript presented in an intelligible fashion and written in standard English?

Reviewer #1: Yes

Reviewer #2: No

5. Review Comments to the Author

Reviewer #1: Manuscript Number: PONE-D-21-29287

Title: Development and real-life use assessment of a self-management smartphone application for patients with inflammatory arthritis. A user-centred step-by-step approach.

Major comments:

The manuscript needs major improvement in reporting the methodology and the abstract as suggested

Other comments:

ABSTRACT: The methods section in the abstract has a lot of missing information about the methodology and data collection process and tools used. Please address following issues in the abstract.

1. Methods: The authors used a ‘mixed-method qualitative–quantitative study’ design but they do not report tools used for the collection of quantitative and qualitative data. It is unclear what the qualitative study involved and what was done in the quantitative part of the study. Please report how did you collect data and which methods and tools/instruments you used for collecting different types of data.

2. Methods: The following sentence is incomplete as it reports two numbers and one category of participants. “A mixed-method qualitative–quantitative study including 42 and 344 patients, respectively”. Please complete the sentence by adding other category of participants.

3. Methods: The authors tested one app; however, they report “the use of health apps”. This needs to be corrected by changing the term ‘apps’ to ‘app’.

4. Methods: The author report “potential use needs”. Is it ‘use needs’ or ‘user needs’?

5. Methods: The author have reported that face to face meetings were held but do not report the number these meeting. Please report the number of meetings held as (n=??).

6. Could the authors add more information about ‘in-depth users’ tests’ because it is unclear what is an in-depth user test.

7. Could the authors report how they collected data on ‘The number of app installations and current users’.

8. It is unclear what is meant by ‘comments at stores’. What is meant by the term ‘stores’ here? Are these shops, super stores or Apple store or google play?

9. Please spell out what ‘HP’ stands for.

10. The author report the name of the app i.e. Hiboot in the results; however, it should have been reported earlier either in the objective or methods sections in the abstract.

INTRODUCTION

1. Could the authors add generic names of a few Disease-modifying anti-rheumatic drugs (DMARDs) in the first paragraph in this section.

2. The authors report that most of the IA apps are developed without input/involvement of patients but they cite only two apps (9,15) out of 7 apps (9-15) reported in the literature in para 2. So please amend your statement claiming: “Most of these apps have been designed without including patients in their development”.

3. The authors report that “The development was designed to (1) involve patients and HPs at every stage,..”. Could you please report the key stages of the development process such as the concept development, design stage, etc…

METHODS

1. Please change the term ‘construction’ to ‘development’ because you were not constructing a building but developing an app.

2. Step 1. Mixed-method qualitative–quantitative study: Please report how rheumatologists recruited patients. How many rheumatologists were involved in the patient recruitment and what methods they used to invite and recruit patients.

3. Step 1. Mixed-method qualitative–quantitative study: Please report how many public hospitals were involved in recruiting patients.

4. Age: Please report which age criteria were used for recruiting patients.

5. Please report the sampling methodology used for the recruitment of study participants.

6. Please add more information about every profiling variable i.e. age, sex, socio-professional status, type of IA, type of DMARDs and number of previous DMARDs. This is because it is unclear what was included and what was excluded with regard to above variables.

7. The authors report that ‘Patient enrolment was stopped at saturation.’ But is unclear how saturation was determined. What was done that indicated saturation?

8. Quantitative survey: The authors conducted an online quantitative survey but it is unclear whether the authors adapted an existing survey or they developed their own survey. The authors need to report information about the survey content (broad areas / constructs covered), validation and piloting process and scoring of items of the survey.

9. Quantitative survey: It was posted on patient association websites and Facebook accounts. Could the authors report the number of the websites and accounts on the Facebook used for this purpose? As well as how many rheumatologists were also involved in this activity.

10. Please report how many patients invited to the survey, how many responses were received, how many incomplete surveys were deleted/excluded from the analysis and how many completed surveys were included in the final data analysis.

11. Please report whether there were missing data and how it was dealt with.

12. The authors used ‘parametric or nonparametric tests’ but they need to be specific, and report / name parametric or nonparametric tests used.

13. Step3. Launch of the first version of the app: Could the authors report key elements of the ‘multimodal strategy’ used for the launch of the app.

14. Step 4: Users’ tests and construction of the current version: Interviews were conducted at this stage. Please report what was the purpose the interviews and which areas were covered in the interview guide and what was the duration of these interviews

15. Step 4: Users’ tests and construction of the current version: Please report how many patients were users and how many non-users of the app among total 7 patients interviewed.

16. Step 4: Users’ tests and construction of the current version: Please report the duration, methods of recording of these interviews with patients as well as rheumatologists.

17. Step 5 (second qualitative study): Please report who was interviewed at this stage: How many patients and rheumatologist, if any, were interviewed at this stage? What was the duration of these interviews?

18. Please report methods used to analyses interview data collected at all steps.

19. Step 6. Current version delivery: Please could you outline which additional features and content were added to Hiboot+ version.

20. App assessment: Could you please report how did you collect data on various items reported in this section? And how these data were analysed.

21. Please when was the app launched? The first version as well as the revised version.

RESULTS:

1. Table 1 and Table 2: The headings of this table may be revised: Change ‘n*’ to ‘total respondents’ and change ‘Results’ to ‘Count (%).

2. Table 1: It is unclear what is the value of reporting the size of place of residence (No. of inhabitants)?

3. Table 2. This table includes difficulties encountered and information needs but it is unclear whether these were related to the use of the app or in the daily life.

4. Results showed that the frequent users were residents of medium sized cities. This is an interesting findings. Could the authors add their reflections on this issue in the discussion section.

5. The authors have reported a lot of text covering information about various healthcare organisations, agencies and professional associations like the Club Rhumatismes et Inflammations (CRI), which might be omitted or reduced because the focus of the study is on the App not the organisations. This can help to reduce the length of the article, which is too long.

6. Please spell out acronym ‘MARS’

Reviewer #2: I read and appreciated the article titled the “Development and real-life use assessment of a self-management smartphone application for patients with inflammatory arthritis. A user-centred step-by-step approach”.

Overall this qualitative study is interesting and it proposed a free to use tool to support patients self-management of inflammatory arthritis.

Nevertheless, there are some issues.

- Often it is not completely clear to me what you want to say, I believe that the text needs some proofreading on certain points, especially the introduction and the methods section are rich but not always to the point. For instance, in the first paragraph of the introduction (last sentence), you say <<they are="" availability="" in="" increasing="">15 DMARDs in France) and have a wide variety of targets and modes of administration>>. At this point I was lost, are you talking about mobile apps?

- The abstract is quite vague, the methods applied are not fairly represented

- The methodology is rich, but not always detailed enough. Moreover, a set of interviews with a small group of patients and experts is presented as a phase of users testing. This can not be right as a user test usually include a usability assessment. Maybe this was a phase of product review, certainly not of testing, or more details should be provided.

- I do not understand the term construction, maybe the authors want to say something like “design”?

- The section App assessment is quite unclear to me. is the author proposing that the number of installations, number of users and type of requests are relevant data? if yes relevant for what? Moreover, it is hard to follow when you are talking about “scores at the store”

- I was wondering if it would be more clear for a reader to distinguish between a qualitative phase of review and redesign of the App and a survey study, instead of between qualitative and quantitative methods as proposed by the authors

Overall I believe that the proposal of this article has merits but the text should be reviewed to make the text more coherent and simple to read, I was between a minor and a major revision but I am inclined to give a minor, by strongly encouraging the authors to reflect on the organization and presentation of the text.</they>

6. PLOS authors have the option to publish the peer review history of their article (what does this mean?). If published, this will include your full peer review and any attached files.

Reviewer #1: **Yes: **Dr Syed Ghulam Sarwar Shah

Reviewer #2: No

---

## [Author Response · Author response to Decision Letter 0]

24 Jan 2022

REPONSE TO REVIEWERS 

We thank the reviewers for their valuable comments that will significantly improve the manuscript. Here are our responses to the reviewers’ comments. We have revised the manuscript accordingly. 

The quoted lines refer to the revised version with track changes. 

Response to reviewer #1

ABSTRACT: The methods section in the abstract has a lot of missing information about the methodology and data collection process and tools used. Please address following issues in the abstract.

1. Methods: The authors used a ‘mixed-method qualitative–quantitative study’ design but they do not report tools used for the collection of quantitative and qualitative data. It is unclear what the qualitative study involved and what was done in the quantitative part of the study. Please report how did you collect data and which methods and tools/instruments you used for collecting different types of data.

Response. We thank the reviewer for this comment and agree that the methods were not clearly indicated in the abstract. Due to the limited length of the abstract (300 words), we were unable to provide sufficient information.

Before the development of the first version of the app, we conducted a first qualitative study with semi-guided audiotaped interviews of 21 patients and a cross- sectional online survey including 344 patients. After the app was launched, we performed a second qualitative study of 21 patients and a users’ test of 13 patients and 3 rheumatologists. Added lines 56-62 in the abstract and 188-205 in the methods. 

In total, the two quantitative studies involved 42 patients. Data were analysed with qualitative methods (detailed in the manuscript). 

We have modified the abstract organisation to give more space to the description of the methods. 

2. Methods: The following sentence is incomplete as it reports two numbers and one category of participants. “A mixed-method qualitative–quantitative study including 42 and 344 patients, respectively”. Please complete the sentence by adding other category of participants. 

Response : There were 42 patients in the two qualitative studies and 344 patients in the quantitative study. 

3. Methods: The authors tested one app; however, they report “the use of health apps”. This needs to be corrected by changing the term ‘apps’ to ‘app’. 

Response: The cross-sectional study investigates the use of health apps in general. The abstract has been modified according to this comment line 59. 

4. Methods: The author report “potential use needs”. Is it ‘use needs’ or ‘user needs’? 

Response : We apologize for this typo that the reviewer has quoted. We meant “user needs”. 

5. Methods: The author have reported that face to face meetings were held but do not report the number these meeting. Please report the number of meetings held as (n=??). 

Response: there were 5 face-to-face meetings, each of them lasting 3 to 4 hours. This was added in the abstract line 60 and methods line 191.

6. Could the authors add more information about ‘in-depth users’ tests’ because it is unclear what is an in-depth user test. 

Response: we thank the reviewer for this comment because it was not clear enough. In-depth user tests consisted in interviews of 1 hour or more exploring the individuals’ use of the app and the users’ feedback. Due to the limited length of the abstract, we have removed the term “in-depth” from the abstract. Details are provided in the manuscript. 

7. Could the authors report how they collected data on ‘The number of app installations and current users’.

Response. We thank the reviewer for this important comment. The details were missing in the abstract and the methods.

The number of app installations and active users was provided by the two mobile phone app stores: Google play store for Android operating systems (https://play.google.com/console/) and Apple app store for iOS (https://appstoreconnect.apple.com/login). These two stores provide analytics features to monitor the performance of apps. The number of installations was defined as the monthly installations minus the monthly uninstallations.

The number of current users was obtained from the mobile phone app stores by counting the monthly number of active devices defined as a unique device on which the app was used at least once within the last 30 days. 

This was added in the abstract, line 63 and in the methods, line 226-231. 

8. It is unclear what is meant by ‘comments at stores’. What is meant by the term ‘stores’ here? Are these shops, super stores or Apple store or google play?

Response. Yes, the comments were collected on Apple store or Google play. 

9. Please spell out what ‘HP’ stands for.

Response. We apologize for this. HP stands for health professionals (HP was removed from the abstract).

10. The author report the name of the app i.e. Hiboot in the results; however, it should have been reported earlier either in the objective or methods sections in the abstract.

Response. We understand this comment as it would have been interesting to disclose the name of Hiboot earlier. This was added in the abstract, line 54 and the introduction, line 124. 

INTRODUCTION

1. Could the authors add generic names of a few Disease-modifying anti-rheumatic drugs (DMARDs) in the first paragraph in this section. 

Response: DMARDs include methotrexate, which is the first-line treatment, biologics such anti-TNF agents, anti IL6, anti-CD 20 etc.. and JAK inhibitors. We did not add generic names but names of the main types of DMARDs: methotrexate, biologics and JAK inhibitors. This was added in lines 93-94. 

2. The authors report that most of the IA apps are developed without input/involvement of patients but they cite only two apps (9,15) out of 7 apps (9-15) reported in the literature in para 2. So please amend your statement claiming: “Most of these apps have been designed without including patients in their development”. 

Response: we thank the reviewer because the references were not clearly described: references 9 and 15 are for 2 systematic reviews. The first one (ref 9) found 17 apps: 11 for RA, 1 for SpA, 1 for inflammatory arthritis and 4 for juvenile idiopathic arthritis. The second review (ref 10) found 6 apps in German, 3 for RA and 3 for SpA. Ref 10 is also a systematic review for 20 apps for monitoring symptoms in RA. 

We have amended the statement as follows “Systematic reviews of IA apps have shown that most apps have been designed without including patients in their development and that healthcare providers have rarely contributed [9,10,15]”. Lines 99-101.

3. The authors report that “The development was designed to (1) involve patients and HPs at every stage,..”. Could you please report the key stages of the development process such as the concept development, design stage, etc…. 

Response: indeed, the 3 key stages of development do not clearly appear. We changed the presentation of the key stages by deleting the numbers (1 to 3) as follows. Line 125-126: “The development was designed to involve patients and HPs, including the development concept, preliminary studies to explore patients' needs and understand the overall aspects of their daily lives, and the use of a step-by-step approach by adjusting the app in line with users’ feedback.”

METHODS

1. Please change the term ‘construction’ to ‘development’ because you were not constructing a building but developing an app. 

Response: We apologize for this. This was corrected in the whole manuscript. 

2. Step 1. Mixed-method qualitative–quantitative study: Please report how rheumatologists recruited patients. How many rheumatologists were involved in the patient recruitment and what methods they used to invite and recruit patients. 

Response: Recruitment for the qualitative studies was conducted in 2 ways. Before starting research, we used a purposeful sampling method to select the participant sampling criteria (please, look at the response to question 6). Then, after the interviews had begun, the sample was completed along with the progression of the data collection (according to the grounded theory model). This was added in line143-149.

The recruitment included an information letter explaining the aim of the study “This study aims to better understand your daily life with your treatment and will be used to offer patients support to help them better manage their disease and their treatment. More specifically, it will be used to develop and improve the French Rheumatology Society's digital tool Hiboot, a free smartphone app for patients with inflammatory arthritis”. 

The 3 rheumatologists of the steering committee were involved in the recruitment. 

We added some of these details in the methods, line 141. 

3. Step 1. Mixed-method qualitative–quantitative study: Please report how many public hospitals were involved in recruiting patients. 

Response: 2 public university hospital recruited the patients. This was added in the methods. Line 142.

4. Age: Please report which age criteria were used for recruiting patients. 

Response: only adult patients were to be recruited with no maximum age. The youngest patient was 17 years old and the oldest 82 years old (please see eAppendix1.2. Participants’ demographics and clinical features). 

5. Please report the sampling methodology used for the recruitment of study participants. 

Response: please see the response to question 1

6. Please add more information about every profiling variable i.e. age, sex, socio-professional status, type of IA, type of DMARDs and number of previous DMARDs. This is because it is unclear what was included and what was excluded with regard to above variables. 

Response: indeed, we did not make this clear enough. 

Sampling criteria were defined according to a purposeful sampling method. It consists of identifying a priori criteria that were important for the subject to be studied.(1)(2) We aimed to recruit a wide range of profiles to obtain the maximum information, so none of the variables were excluded. Patients had to be recruited by age, sex and disease duration. They had to belong to various socio-professional groups. Recruitment also involved patients with different types of IA rheumatoid arthritis, spondyloarthritis, psoriatic arthritis), with different types of DMARDs (methotrexate only, biologics etc..) and who had received a various number of DMARDs. Please see eAppendix1.2. Participants’ demographics and clinical features). 

We added some of this additional information in line 145. 

We have detailed inclusion and exclusion criteria in the methods, as follows “The inclusion criteria were adult patients, with a diagnosis of IA confirmed by the rheumatologist according to international diagnostic criteria who received at least one DMARD, were followed as outpatients or inpatients, and were fluent in French. Exclusion criteria were conditions that could alter the patients’ understanding such as cognitive impairment and psychiatric disorders.” Line 152-156.

(1) Palinkas LA, Horwitz SM, Green CA, Wisdom JP, Duan N, Hoagwood K. Purposeful Sampling for Qualitative Data Collection and Analysis in Mixed Method Implementation Research. Administration and Policy in Mental Health and Mental Health Services Research. 1 sept 2015;42(5):533‑44. 

(2) Marshall MN. Sampling for qualitative research. Family Practice. 1 jan 1996;13(6):522‑6. 

7. The authors report that ‘Patient enrolment was stopped at saturation.’ But is unclear how saturation was determined. What was done that indicated saturation?

Response. Saturation is achieved when adding more participants does not provide any new information (added line 163-164.) Saturation is often observed with a small number of participants (about 10), but the determination of the saturation point is not based on a standard calculation and depends on the experience of the researchers. 

In each study, a saturation point was estimated a priori: according to the recruitment criteria, we estimated that we needed to interview at least 20 patients. In both studies, saturation was achieved. 

Refs : Guest, G., Bunce, A., & Johnson, L. (2006). How Many Interviews Are Enough?: An Experiment with Data Saturation and Variability. Field Methods, 18(1), 59–82. 

Sandelowski, M. (1995), Sample size in qualitative research. Res. Nurs. Health, 18: 179-183. 

8. Quantitative survey: The authors conducted an online quantitative survey but it is unclear whether the authors adapted an existing survey or they developed their own survey. The authors need to report information about the survey content (broad areas / constructs covered), validation and piloting process and scoring of items of the survey.

Response. We agree that information is missing on these points. The survey questionnaire was developed by the steering committee based on the study purpose. The survey content was drawn from the qualitative study with a focus on patients’ practices with their treatments and the use of apps. 

The broad areas of the survey were demographics, difficulties and problems encountered in daily life, needs for advice, current use of apps and health apps, potential use of apps for IA and type of situations where an app would be useful, and unrestricted opinion on app use. 

More details on the development and main areas of the survey have been added in lines 171-173.

The scoring of items depended on the type of question: either yes/no answers or answers on a Likert scale. The types of answers are presented in Tables 2 and 3. 

9. Quantitative survey: It was posted on patient association websites and Facebook accounts. Could the authors report the number of the websites and accounts on the Facebook used for this purpose? As well as how many rheumatologists were also involved in this activity. 

Response. Each patient association had its own website and Facebook account. Three partner associations participated (added in the methods line 173). No rheumatologist Facebook accounts were used. 

Rheumatologists anonymously proposed the survey to their patients and provided the Internet link to patients (line 178). The rheumatologists involved belonged to the 2 university rheumatology departments and the 2 private offices added in line 178. However, because of the anonymous data collection, we were not able to determine the number of rheumatologists who actually participated in the patients’ enrolment. 

10. Please report how many patients invited to the survey, how many responses were received, how many incomplete surveys were deleted/excluded from the analysis and how many completed surveys were included in the final data analysis.

Response. The patients invited in the survey were all patients belonging to the 3 partner patient associations and the patients anonymously recruited by the rheumatologists. Because of the anonymous data collection, we were not able to determine the number of patients invited to the survey. For further response, please look at responses to question 11. 

11. Please report whether there were missing data and how it was dealt with.

Response. Yes, this is important information available in line 297: the quantitative study included 344 patients, and 331 questionnaires were complete for analysis. The number of data available are presented in Tables 1 to 3, which shows the number of missing data. Of note, the number of incomplete questionnaires was low (3.7%). Missing data were not imputed because this is a descriptive study. Reporting the number of available data and the descriptive analysis of available data was not appropriate in this context. 

12. The authors used ‘parametric or nonparametric tests’ but they need to be specific, and report / name parametric or nonparametric tests used.

Response. This information was missing. According to the statistical distribution, the parametric tests were chi-square test and nonparametric Fisher’s exact test for categorical variables and Wilcoxon rank test for quantitative variables. This was added in the methods lines 186-188 and the results lines 332-334. 

13. Step3. Launch of the first version of the app: Could the authors report key elements of the ‘multimodal strategy’ used for the launch of the app.

Response. Yes, this is an important comment. In the methods, we stated that the launch included communication to patients, rheumatologists and health professionals, lines 195-196. The details are in the results, lines 432-437, as follows: “The launch campaign (Step 3) included emails to the rheumatologists on the French Society of Rheumatology mailing list, monthly newsletters from the CRI network, and printed flyers and posters for outpatient clinics in private rheumatology offices and departments. Advertising was produced for the front page of the French Society of Rheumatology website and for social networks (Facebook, Twitter @AppliHiboot) as well as for patient association websites and Facebook/Twitter accounts. A video was posted on YouTube. “

14. Step 4: Users’ tests and construction of the current version: Interviews were conducted at this stage. Please report what was the purpose the interviews and which areas were covered in the interview guide and what was the duration of these interviews

Response. We thank the reviewer for this comment because although the results of the user’s tests are detailed in eAppendix 1 to 4, multimedia appendix 3, we did not give enough information in the methods and the results. 

In the methods, we changed the paragraph as follows: line 198-212 “Two rounds of user tests were conducted. The first round in September 2018 consisted of in-depth interviews with 7 patients of different profiles (current users or those who dropped out) and 3 rheumatologists who recommended or not the app to their patients. The second round of user tests conducted in December 2019 included 6 patients, 4 recruited by rheumatologists and 2 recruited by Stephenson, an agency specialized in market analysis, from a panel of volunteers. The aim was to collect patients' opinions on the app's features, interface, and navigation and to investigate the opinions of rheumatologists. Patient interviews took place in a neutral environment (coffee shop, workplace) or by telephone. Interviews with rheumatologists were by telephone. Interviews were conducted according to a predefined schedule. The main topics for the patient schedule were their comments on the use and functionality of the app. The main topics for the rheumatologist interviews were their own knowledge of the app, their own feedback on the app and patient feedback, and barriers to recommending the app.” 

In the results, we added some data drawn from the appendix 3, Lines 442-444: “In total 13 patients and 3 rheumatologists were interviewed. Patient interviews lasted from 90 to 120 min and rheumatologist interviews from 30 min to 1 hour”. 

15. Step 4: Users’ tests and construction of the current version: Please report how many patients were users and how many non-users of the app among total 7 patients interviewed.

Response. Among the 13 interviewed patients (7+6), 7 were using the app, 2 had dropped out and 4 were non-users (appendix 3). Added in the manuscript, lineS 442-443. 

16. Step 4: Users’ tests and construction of the current version: Please report the duration, methods of recording of these interviews with patients as well as rheumatologists.

Response. Please see the response to questions 14 and 15. 

17. Step 5 (second qualitative study): Please report who was interviewed at this stage: How many patients and rheumatologist, if any, were interviewed at this stage? What was the duration of these interviews? 

Response. We agree that this point was not explain clearly enough. Only patients were interviewed in this step. The duration of interviews and methods were the same as in step 1. Added as follows: “Recruitment sampling, data collection and data analysis were conducted using the same methods as in Step 1.” line 220-221

18. Please report methods used to analyses interview data collected at all steps.

Response. Information on data analysis is reported in lines 160-164 for the two qualitative studies. 

19. Step 6. Current version delivery: Please could you outline which additional features and content were added to Hiboot+ version.

Response. The additional features and content are detailed in the results, lines 456- 459, and in Text box 2. To avoid redundancy, we have not detailed in the methods the changes between the first version and the current version.

20. App assessment: Could you please report how did you collect data on various items reported in this section? And how these data were analysed.

Response. We assessed the app with 3 metrics that could easily be collected by using analytics features provided by the app stores (the platform where users can download the apps from Apple and Google): number of downloads, monthly active users and users’ evaluations (5-star rating). These metrics are provided by the mobile app stores (Apple and Google). This was added in the methods, line 226-229.

21. Please when was the app launched? The first version as well as the revised version.

Response. The first version was launched in May 2017, with small changes (bug corrections) made until January 2018 and the current version was launched in December 2019 on iOs and January 2020 on Android, with bug corrections made until July 2020 (Figure 1). Added in the manuscript in the methods, lines 196 and 224, and in the results, line 437-438 and 454-455.

RESULTS:

1. Table 1 and Table 2: The headings of this table may be revised: Change ‘n*’ to ‘total respondents’ and change ‘Results’ to ‘Count (%). Response. This was corrected. 

2. Table 1: It is unclear what is the value of reporting the size of place of residence (No. of inhabitants)? Response. Yes it was the No. of inhabitants

3. Table 2. This table includes difficulties encountered and information needs but it is unclear whether these were related to the use of the app or in the daily life.

Response. The difficulties encountered and information needs are those reported in daily life. Added in the table. 

4. Results showed that the frequent users were residents of medium sized cities. This is an interesting findings. Could the authors add their reflections on this issue in the discussion section.

Response. We thank the reviewer for quoting this interesting point. We were not able to compare this point with other studies because to our knowledge, this information was not available in published surveys. One can deduce that individuals living in medium-size cities may have less access to a rheumatologist and need more counselling besides the traditional health care system. Because of the length of the manuscript and the fact that this result was only found in our survey, we chose not to discuss this point. Another study is planned to assess Hiboot’s benefit. We will also include this factor in the demographics characteristics. 

5. The authors have reported a lot of text covering information about various healthcare organisations, agencies and professional associations like the Club Rhumatismes et Inflammations (CRI), which might be omitted or reduced because the focus of the study is on the App not the organisations. This can help to reduce the length of the article, which is too long.

Response. We agree with the reviewer’s concern about the article length. We believed that these details were necessary because the app deals with safety issues. We intended to show the reliability of the app’s content in this regard. To respond to the reviewer comments, we have reduced this part in the results and created an additional eAppendix in the Multimedia appendix 2 (Hiboot Features and content examples), named “Hiboot information sources” where we detailed the healthcare organisations, agencies and professional associations that provided the scientific content of the safety messages in daily life. 

6. Please spell out acronym ‘MARS’

Response. We apologise for this omission. The MARS is the Mobile Application Rating Scale. Line 570 In the discussion, we have changed one sentence about the MARS because a user MARS version is available: line 576 ref 61, was added. 

Reviewer #2: I read and appreciated the article titled the “Development and real-life use assessment of a self-management smartphone application for patients with inflammatory arthritis. A user-centred step-by-step approach”.

Overall this qualitative study is interesting and it proposed a free to use tool to support patients self-management of inflammatory arthritis.

Nevertheless, there are some issues.

- Often it is not completely clear to me what you want to say, I believe that the text needs some proofreading on certain points, especially the introduction and the methods section are rich but not always to the point. For instance, in the first paragraph of the introduction (last sentence), you say <15 DMARDs in France) and have a wide variety of targets and modes of administration>>. At this point I was lost, are you talking about mobile apps?

Response. We thank the reviewer for the comments. We believe that there is a misunderstanding on the last sentence of the introduction. We stated that more than 15 DMARDs were available in France. This is one of the rationales for developing an app for self-management because managing medications in daily life is part of self-management. We changed the sentence as follows “(more than 15 DMARDs in France)” line 94-95. 

- The abstract is quite vague, the methods applied are not fairly represented

Response. We agree with the reviewer. Because of the limited length of the abstract (300 words), we were not able to provide as sufficient information to make the methods clear. We have changed the abstract organisation to give more space to the description of the methods. 

- The methodology is rich, but not always detailed enough. Moreover, a set of interviews with a small group of patients and experts is presented as a phase of users testing. This can not be right as a user test usually include a usability assessment. Maybe this was a phase of product review, certainly not of testing, or more details should be provided.

Response. We thank the reviewer for this valuable comment. There were two kinds of interviews. The interviews for the 2 qualitative studies were conducted according the guidelines (COREQ in supplementary material 1). The user tests, as the reviewer underlined, aimed to collect the opinion of patients and rheumatologists on the usability of the app. Details have been provided in the manuscript, lines 198-211, in the methods section.

- I do not understand the term construction, maybe the authors want to say something like “design”?

Response. We agree with the reviewer’s comment. The term construction has been replaced by “development” as suggested by reviewer #1

- The section App assessment is quite unclear to me. is the author proposing that the number of installations, number of users and type of requests are relevant data? if yes relevant for what? Moreover, it is hard to follow when you are talking about “scores at the store”

Response. We thank the reviewer for raising this point, which has rarely been addressed in the literature on health applications. There is no literature on relevant criteria for assessing mobile health applications. The Mobile Application Rating Scale (MARS) is a questionnaire developed to evaluate mhealth applications from the perspective of users or health professionals. (cited in the discussion). However, some systematic reviews of mobile health applications in rheumatology have shown that, although the design of applications is well described and sometimes assessed, most applications were not available after their launch or had a very low usage rate. Indeed, the number of users is rarely reported. We thought that this information could be a marker of the usefulness of the application. Therefore, we found it relevant to collect the number of installations, the number of users and the type of requests, which meant that patients found the application useful in their daily life, although we could not provide references on this statement. In this respect, Hiboot has a longer lifespan than most other applications in the literature. Nevertheless, as mentioned in the discussion, these data are not sufficient to evaluate Hiboot, and further studies are needed. 

Indeed, the “scores at stores” is not clear enough. We meant on the app stores (Apple store and Google play). The scores can be consulted before downloading the app. The scores are determined by the app’s users, rated from 1 star to 5 stars. Users can also post comments. The full analysis of these comments has been carried out in this study. This was added in the abstract, lines 63, and the methods, lines 226-229. 

- I was wondering if it would be more clear for a reader to distinguish between a qualitative phase of review and redesign of the App and a survey study, instead of between qualitative and quantitative methods as proposed by the authors

Response. We apologise for the misunderstanding regarding the development stages and agree that the development may be hard to follow. Indeed, the development took quite some time between the preliminary stages in May-June 2016 and the release of the current version in December 2019. The first qualitative study and the survey were part of the preliminary studies before the development of the app and the second qualitative survey took place after the app launch. In accordance with the reviewer’s comment, these 2 studies are reported together in the results. 

Only the users’ tests are reported separately because their aims and methods were different. 

For the rest of the development, a chronological report was chosen. We hope that Figure 1 can help the reader understand the different stages.

---

## [Decision Letter · Decision Letter 1]

16 Feb 2022

PONE-D-21-29287R1Development and real-life use assessment of a self-management smartphone application for patients with inflammatory arthritis. A user-centred step-by-step approach.PLOS ONE

Dear Dr. Beauvais,

Thank you for submitting your manuscript to PLOS ONE. After careful consideration, we feel that it has merit but does not fully meet PLOS ONE’s publication criteria as it currently stands. Therefore, we invite you to submit a revised version of the manuscript that addresses the points raised during the review process.

We look forward to receiving your revised manuscript.

Kind regards,

Jianhong Zhou

Associate Editor

PLOS ONE

Journal Requirements:

Reviewers' comments:

Reviewer's Responses to Questions

**Comments to the Author**

1. If the authors have adequately addressed your comments raised in a previous round of review and you feel that this manuscript is now acceptable for publication, you may indicate that here to bypass the “Comments to the Author” section, enter your conflict of interest statement in the “Confidential to Editor” section, and submit your "Accept" recommendation.

Reviewer #1: (No Response)

Reviewer #2: All comments have been addressed

2. Is the manuscript technically sound, and do the data support the conclusions?

Reviewer #1: Yes

Reviewer #2: Yes

3. Has the statistical analysis been performed appropriately and rigorously? 

Reviewer #1: No

Reviewer #2: Yes

4. Have the authors made all data underlying the findings in their manuscript fully available?

Reviewer #1: No

Reviewer #2: Yes

5. Is the manuscript presented in an intelligible fashion and written in standard English?

Reviewer #1: No

Reviewer #2: Yes

6. Review Comments to the Author

Reviewer #1: The authors have submitted the revised manuscript entitled “Development and real-life use assessment of a self-management smartphone application for patients with inflammatory arthritis. A user-centred step-by-step approach’, which is much improved compared to the original version. However, the manuscript still has some issues that need to be addressed by the authors as suggested below.

Major comments

1. The conclusion reported in the abstract and the main body of manuscript is more like a commercial and selling comments rather than a scientific remarks in line with the major findings of the research study.

2. The authors report that they used a parametric technique i.e. chi squared test (line 186, page 9), which is incorrect. The chi squatted test is a non-parametric test, please correct the text.

3. Results Section Summary of the key emerging themes (Lines 248-289). In this section the authors report only names of themes and sub-themes without any direct quotes of patients. The authors need to report some representative quotes for each theme in this section.

4. Please delete ‘eAppendix’ throughout the text because it is mentioned as either ‘multimedia’ or ‘supplementary material’. Please use only one term for the supplementary material and double check the content and number of each of the supplementary material file.

Minor comments

ABSTRACT

5. Line 77: Could you please report any specific dates/months in the reported period i.e. 2017-2020.

KEYWORDS

6. Line 83: It would be better Not to report an acronym i.e. DMARDs as a keyword but spell it out and report it as ‘Disease-modifying antirheumatic drugs (DMARDs)’

INTRODUCTION

7. Line 110: Please be specific and report whose education (patients or users) and about what are you referring to the following sentence “Few apps for IA patients have targeted education [9,.10,.15].”

8. Line 126: Please delete 'Comments: OK'

9. Line 129: Please add 'app’ before the term ‘stores’.

METHODS

10. Line 142: Please check and change ‘private office’ to ‘private clinics/hospitals’ as appropriate.

11. Line 149: The authors report using ‘the grounded theory model’. Could you please provide a reference/citation about the 'grounded theory model' used?

12. Line 152: Please report what age was considered as ‘adults’. You have mentioned in the side comments the age of ≥ 18 years.

13. Lines 186-188: All these statistical tests are non-parametric tests. Please revise the statement and report the techniques used and the types of data analysed by the tests.

14. Line 197-200: Step 4 User tests: 6 patients recruited (current users and dropped out) in the 1st round (Sep 2018). Please report the number of current users and dropped outs users included in this stage

15. Lines 203-205: In the 2nd round (Dec 2019): 6 patients were also recruited. Could you please report whether all of them were current users or included some dropped outs users. if so please, report their number in each category.

16. Lines 214: The authors report ‘…added 2 years after the app’s launch..”. Please report the months/year during which the 2nd qualitative study was undertaken so that a clear timeline is available for the readers.

17. Line 235: The authors harnessed app users’ comments. Could you please how many unique users' comments were collected and analysed.

18. Were your patients the same people in the first and second qualitative studies or different people in each study?

19. The authors did different studies over 4-5 years. Could you please sign post the research activities and timeline reported in figure 1 in the text.

RESULTS

20. Lines 240-244: The authors report that there were 21 patients each in study 1 and study 2 so the total patients become 42. However, the authors report female patients as 33/41, which should be 33/42.

21. Please correct the total of your patients (n=42) in the following sentence, which shows total 41 = (11+30) patients.

22. Lines 244-246: The authors report that "Eleven received methotrexate monotherapy and 30 bDMARDs or tsDMARDs monotherapy." So there were 11+30=41 patients. Please double check whether you had 42 or 41 total patients in the studies.

23. Lines 248- Section Summary of the key emerging themes. In this section the authors report only names of themes and sub-themes without any direct quotes of patients. The authors need to report some representative quotes for each theme in this section.

24. Line 383: keywords: Could you please signpost the readers which keywords and where they are reported in the manuscript.

25. Lines 480: Please double check whether the following is correct: views/month/users. It could be views/month/user. (Such as XX views per month per user (NOT users)).

26. Lines 285-590: Please report what numbers given in parenthesis in these sentences show? These number could be confused and read as reference / citations.

DISCUSSION

27. Line 500: the authors report ‘…..4300 users so far.’ Could you please the exact date on which this number was ascertained?

28. Line 501: Preliminary studies: Please double check whether you mean Preliminary studies or previous/earlier studies. The later seems more appropriate in the context of the sentence. Please check whether this needs to be changed elsewhere in the manuscript.

29. Line 518: The authors report that “To our knowledge, no app of this nature is currently available for adult patients with IA.”. Again, in lines 551-552, the authors talk about other apps for patients with IA and state “However, this number seems substantial as compared with other apps for people with IA”. Please double check the contradictory statements and revise as appropriate.

CONCLUSION

30. The conclusion is more like a commercial and selling statement rather than scientific remarks about the app and the study findings.

Tables and boxes etc.

31. Text box 1 should include a column where the authors should report direct quotes from the participants.

32. Figure 3 caption, Please reports the dates month/years in the caption which only includes ‘since October 2017’ which should be ‘October 2017-Month/Year) to show the period a

Reviewer #2: The adequately addressed my comments . They explained more in details the methodology and reviewed the language.

7. PLOS authors have the option to publish the peer review history of their article (what does this mean?). If published, this will include your full peer review and any attached files.

Reviewer #1: **Yes: **Syed Ghulam Sarwar Shah

Reviewer #2: No

---

## [Author Response · Author response to Decision Letter 1]

26 Apr 2022

We have reviewed and responded to the reviewer's and editor comments

---

## [Decision Letter · Decision Letter 2]

15 Jul 2022

Development and real-life use assessment of a self-management smartphone application for patients with inflammatory arthritis. A user-centred step-by-step approach.

PONE-D-21-29287R2

Dear Dr. Beauvais,

We’re pleased to inform you that your manuscript has been judged scientifically suitable for publication and will be formally accepted for publication once it meets all outstanding technical requirements.

Kind regards,

Vanessa Carels

Staff Editor

PLOS ONE

Additional Editor Comments (optional):

Reviewers' comments:

Reviewer's Responses to Questions

**Comments to the Author**

1. If the authors have adequately addressed your comments raised in a previous round of review and you feel that this manuscript is now acceptable for publication, you may indicate that here to bypass the “Comments to the Author” section, enter your conflict of interest statement in the “Confidential to Editor” section, and submit your "Accept" recommendation.

Reviewer #1: All comments have been addressed

2. Is the manuscript technically sound, and do the data support the conclusions?

Reviewer #1: Yes

3. Has the statistical analysis been performed appropriately and rigorously? 

Reviewer #1: Yes

4. Have the authors made all data underlying the findings in their manuscript fully available?

Reviewer #1: Yes

5. Is the manuscript presented in an intelligible fashion and written in standard English?

Reviewer #1: Yes

6. Review Comments to the Author

Reviewer #1: The authors have adequately addressed all issues raised in my earlier report. Thanks. The manuscript is clear and improved.

7. PLOS authors have the option to publish the peer review history of their article (what does this mean?). If published, this will include your full peer review and any attached files.

Reviewer #1: **Yes: **Syed Ghulam Sarwar Shah

---

## [Editor Report · Acceptance letter]

4 Sep 2022

PONE-D-21-29287R2 

Development and real-life use assessment of a self-management smartphone application for patients with inflammatory arthritis. A user-centred step-by-step approach. 

Dear Dr. Beauvais:

I'm pleased to inform you that your manuscript has been deemed suitable for publication in PLOS ONE. Congratulations! Your manuscript is now with our production department. 

Kind regards, 

on behalf of

Dr. Vanessa Carels 

Staff Editor

PLOS ONE